# Options for reforming agricultural subsidies from health, climate, and economic perspectives

M. Springmann[1,3 ✉] & F. Freund [1,2,3 ✉]

Agricultural subsidies are an important factor for influencing food production and therefore part of a food system that is seen as neither healthy nor sustainable. Here we analyse options for reforming agricultural subsidies in line with health and climate-change objectives on one side, and economic objectives on the other. Using an integrated modelling framework including economic, environmental, and health assessments, we find that on a global scale several reform options could lead to reductions in greenhouse gas emissions and improvements in population health without reductions in economic welfare. Those include a repurposing of up to half of agricultural subsidies to support the production of foods with beneficial health and environmental characteristics, including fruits, vegetables, and other horticultural products, and combining such repurposing with a more equal distribution of subsidy payments globally. The findings suggest that reforming agricultural subsidy schemes based on health and climate-change objectives can be economically feasible and contribute to transitions towards healthy and sustainable food systems.

[1] Oxford Martin Programme on the Future of Food and Nuffield Department of Population Health, University of Oxford, Old Road Campus Headington, Oxford OX3 7LF, UK. [2] Johann Heinrich von Thünen Institute—Federal Research Institute for Rural Areas, Forestry and Fisheries, Institute of Market Analysis, Bundesallee 63, Braunschweig 38116, Germany. [3]These authors contributed equally: M. Springmann, F. Freund. ✉email: marco.springmann@ndph.ox.ac.uk; florian.freund@thuenen.de

The current food system is neither healthy, nor sustainable. Imbalanced diets, such as diets too low in fruits, vegetables, legumes and nuts, and too high in red and processed meat, are responsible for the greatest mortality burden globally and in most regions[1], and the prevalence of overweight and obesity has increased by over a third in the last 30 years[2]. When it comes to the environment, the food system is responsible for a third of all greenhouse gas (GHG) emissions and therefore a major driver of climate change[3]. It also uses about three quarters of all freshwater resources and occupies more than a third of the Earth's land surface, which puts pressures on ecosystems and biodiversity[4].

Model-based analyses suggest that in addition to technological innovation and changes in farming practices, also large-scale dietary changes and concomitant changes in agricultural production will be needed to achieve healthy diets for a growing population, whilst staying within the environmental limits of the food systems[5]. For example, instead of additional global increases in the production of staple crops, animal-source foods, and sugar crops—estimated at 40–80% between 2010 and 2050—a food system underpinning healthy and sustainable diets would require shifts from those food groups to foods that are both healthy and lower in environmental resource use and pollution, such as fruits, vegetables, legumes, and nuts and seeds.

Reforming agricultural subsidies could play a role in supporting shifts towards healthier and more sustainable food systems. Agricultural subsidies are an important factor for influencing production. In 2016, they represented 25% of the value of production in OECD countries, and 15% in non-OECD countries[6]. Although subsidies have become increasingly decoupled, commodity-specific support measures still represent a significant portion of agricultural subsidies either through direct coupling or through market-price support, and decoupled payments have often supported the continuation of once coupled production systems. The importance of aligning agricultural subsidies with a comprehensive set of societal goals that include both health and the environment is increasingly recognised[7–10], but quantitative analyses that adopt a comprehensive food-systems perspective that goes beyond tracking changes in production are largely lacking.

In this study, we address this gap by constructing an integrated economic-environmental-health modelling framework and using it to analyse options for reforming agricultural subsidies that are in line with health and climate-change objectives. Our analysis shows that on a global scale several reform options could lead to reductions in GHG emissions and improvements in population health without reductions in economic welfare. Those include a repurposing of up to half of agricultural subsidies to support the production of foods with beneficial health and environmental characteristics, including fruits, vegetables, and other horticultural products, and combining such repurposing with a more equal distribution of subsidy payments globally. The findings suggest that reforming agricultural subsidy schemes based on health and climate-change objectives can be economically feasible and contribute to transitions towards healthy and sustainable food systems.

## Results

We used an integrated modelling framework for our analysis. For building the economic-environmental-health modelling framework, we combined a detailed economic representation of agricultural subsidies[11] with region and commodity-specific environmental footprints[5], and with a health assessment of the burden of diet-related diseases that are associated with dietary risk factors, such as low intake of fruits and vegetables, and high intake of red meat[12] ("Methods"). In our environmental analysis,

we focus on changes in agricultural GHG emissions (specifically methane and nitrous oxide) because GHG emissions, compared to other environmental impacts, are less modifiable by farm-level management and more by changes in the mix of production[5]. Within the framework, we account for the dynamic interactions that e.g. changes in diet-related diseases have on the labour force and thus on economic output, and how price and supply-demand reactions influence production, consumption, trade, and the distribution of environmental impacts.

We used the modelling framework to analyse various options for reforming agricultural subsidies in line with health and climate-change objectives. The options we considered ranged from a complete removal of subsidies; over partial and complete coupling of subsidies to food commodities with beneficial environmental and health characteristics; to structural changes in the global subsidy scheme that, in addition to the repurposing of subsidies, included a more equal provision of subsidies across countries. For the coupling of subsidies, we adopted a food-group approach and, in line with projections of the required food-system transformation for healthy and sustainable diets, redirected different proportions of subsidies to the production of horticultural commodities (fruits, vegetables, legumes, and nuts) that have been associated with beneficial health and environmental characteristics.

Our food-group focus on horticultural products can be seen as analogous to approaches that aim to condition subsidies explicitly to the actual health and environmental characteristics of food commodities. Life-cycle analyses indicate that the impacts of what type of food is grown far outweighs how it is grown, especially when comparing animal source foods with plant-based ones, and when comparing different foods within the same region[13,14]. Similarly, epidemiological studies indicate that non-starchy plant-based foods such as fruits, vegetables, legumes, and nuts are associated with reduced risks for various diet-related diseases, while other foods are either associated with increased risk (red and processed meat) or are seen as relatively risk neutral (poultry and dairy) compared to baseline diets[4,15]. Here we focus on these general health and environmental characteristics of horticultural foods, noting that additional differentiation might sometimes be appropriate.

**Current subsidies**. Agricultural support measures, excluding trade tariffs and subsidies, totalled USD 233 billion globally in 2017 (Table S9). More than half (55%) were spent by OECD countries, in particular the EU (32%), USA (12%), and Japan (3%), and much of the remainder (45%) by non-OECD countries, including China (25%) and India (15%). Globally, about 8% of all subsidies were directly coupled to a single output, and the remaining share benefited either special groups of commodities (29%), all commodities without differentiation (31%), or farmers directly without requiring production (31%). Analysed by final use, a fifth to a quarter of agricultural support measures were each used to grow staple crops (22%), meat products (22%), and fruits and vegetables (24%), and about a tenth each for milk and dairy (10%), oil and sugar (12%), and other crops (11%) (Fig. 1 and Supplementary Table 10).

**Removal of subsidies**. Removing all agricultural subsidies by 2030 led to reduced production of crops that were previously supported, including grains, fruits and vegetables, and oil seeds in OECD countries (−1.1 to −2.8% on average), and fruits, vegetables, and milk products in non-OECD countries (−0.8 to −1.2%) (Fig. 2a). Regions that had no subsidies to remove reacted by increasing production (+0.6%), but this did not compensate the reductions in other regions. GHG emissions mirrored the changes in

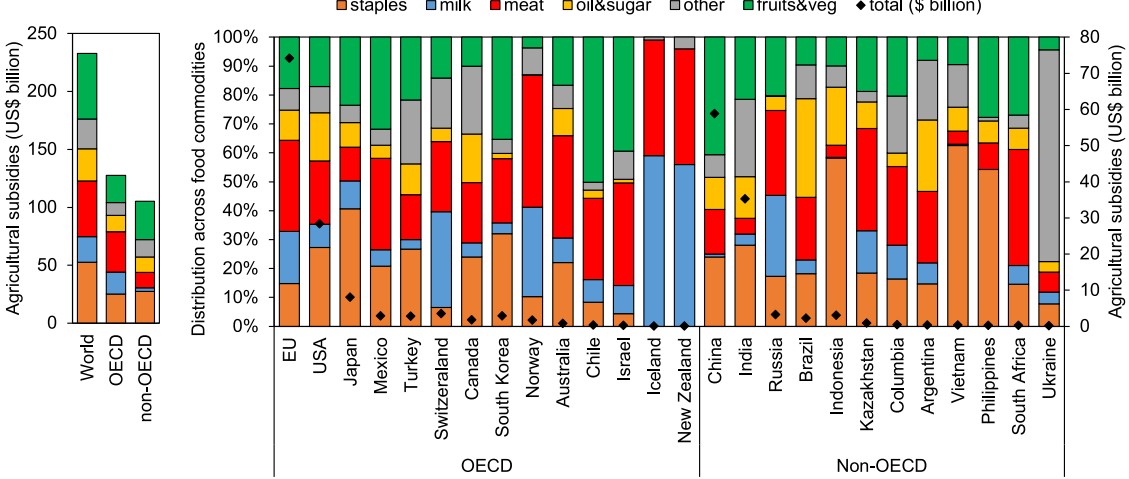

**Fig. 1 Overview of agricultural support measures in 2017, including major spenders and the distribution by final use per commodity.** Total subsidy payments for major spenders, grouped by OECD and non-OECD countries, are shown on the right axis and percentage distribution on the left axis.

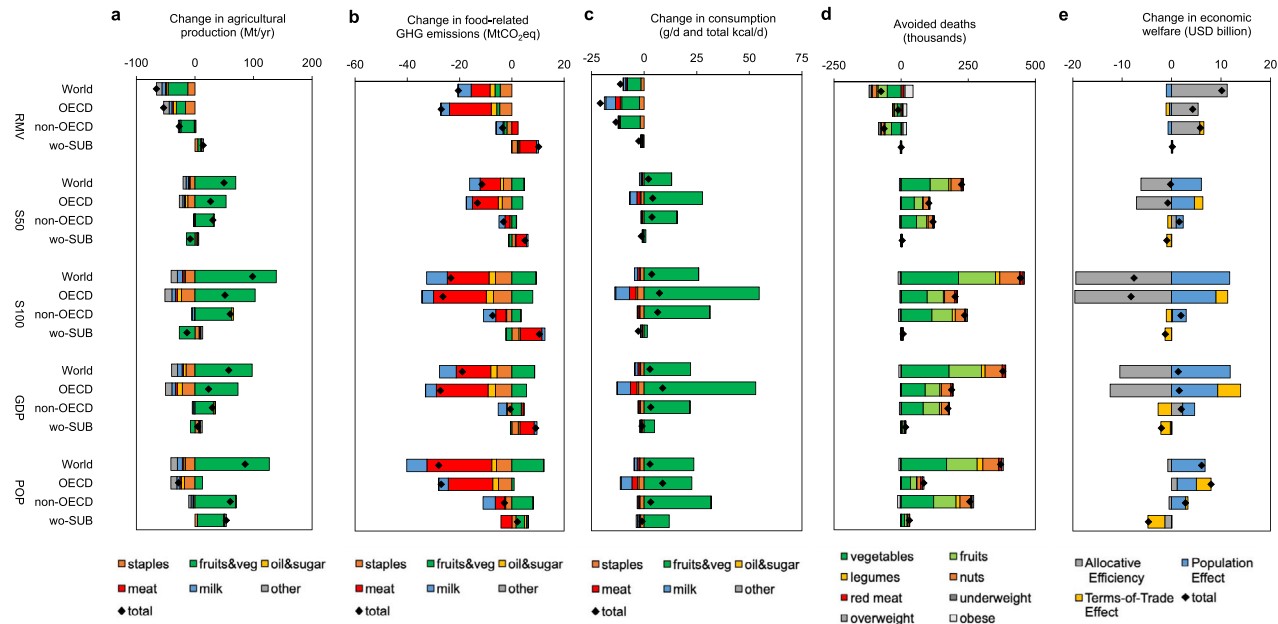

**Fig. 2 Impacts of agricultural subsidy reform.** The impacts include changes in food production (**a**), GHG emissions (**b**), food consumption (**c**), diet-related mortality (**d**), and economic welfare (**e**) by scenario, region, and component. Impacts are for the year 2030 compared to a business-as-usual scenario without reforms. The reform scenarios include a complete removal of agricultural subsidies (RMV), a repurposing of 50% (S50) and 100% (S100) of subsidies for the production of food commodities with beneficial health and environmental characteristics, and a combination of repurposing and regional restructuring in which each country provides subsidies in proportion to either its economy (GDP) or population (POP), whilst keeping the global amount of subsidy payments fixed. Regions include OECD countries with agricultural subsidies (OECD), non-OECD countries with agricultural subsidies (non-OECD), countries without agricultural subsidies (wo-SUB), and a combination of all countries (World). Food groups include wheat, and other cereals and grains (staples), vegetables, fruits, and other horticultural products (fruits&veg), vegetable oils and sugar (oil&sugar), beef, lamb, pork, and poultry (meat), milk and dairy products (milk), and other food commodities (other). Percentage changes and impacts for more specific regions and countries are presented in the Supplementary Information.

production and were moderately reduced in OECD countries (−1.8%), slightly reduced in non-OECD countries (−0.1%), and slightly increased in non-subsidizing countries (+0.5%) (Fig. 2b). Individual countries exhibited larger changes (Supplementary Fig. 5).

The changes in consumption followed the changes in production, but were mediated by changes in trade and commodity prices. The per-capita consumption of fruits, vegetables, and other horticultural products decreased in all regions (6 g/d, 1–9 g/d across regions), as did total energy intake

(11 kcal/d, 2–21 kcal/d across regions) (Fig. 2c). Associated with those changes was a net increase in diet-related mortality (+75,000 deaths in 2030; 95% confidence interval (CI), 71,000−80,000), most of which was associated with the reductions in fruit and vegetable consumption in both OECD and non-OECD countries, but slightly compensated by reductions in overweight and obesity (Fig. 2d).

The increases in mortality affected the labour supply and led to a reduction in economic welfare (measured as equivalent variation in income) of about USD 1 billion (Fig. 2e). However,

the reduction was overcompensated by increases in allocative efficiency associated with a more efficient use of labour outside of the previously subsidised agricultural sectors (USD 11 billion). In addition, changes in the terms of trade played a role for particular regions. For example, associated with the removal of subsidies were increases in world market prices, which decreased the gains from trade (i.e. the terms of trade) for net importing regions such as the OECD (USD 1.1 billion), and increased the terms of trade for net exporting regions. The net economic impact was positive for most regions.

**Repurposing of subsidies**. Using agricultural subsidies to support the production of foods with beneficial health and environmental characteristics (i.e. repurposing from previous ways of allocating subsidies, Fig. 1) led to increased production of horticultural products in OECD countries (+19% for complete repurposing) and non-OECD countries (+3%), and to slight reductions in non-subsidizing countries (−2.4%) (Fig. 2a). GHG emissions were moderately reduced in OECD countries (−1.7%) due to reductions in animal source foods and staple crops that accompanied the increases in horticultural products, they stayed similar in non-OECD countries (−0.2%) that had relatively less subsidies directed towards those crops, and they increased slightly in non-subsidising countries (+0.5%) as they partly compensated the reductions in animal source foods of other countries (Fig. 2b).

The consumption of fruits, vegetables, and other horticultural products increased significantly when all subsidies were repurposed (Fig. 2c). In OECD countries, fruit and vegetable consumption increased by 55 g/d (10%) on average, and in non-OECD countries by 31 g/d (5%). Consumption in non-subsidising countries also increased (2 g/d, 0.3%) as global increases in production reduced global market prices. The changes in consumption, in particular of fruits and vegetables, led to reductions in diet-related mortality that amounted to 444,000 (95% CI, 429,000−460,000) less deaths in 2030 in total, with a similar geographic distribution (Fig. 2d).

In the economic analysis, the reductions in diet-related mortality led to economic benefits associated with an increased labour supply (USD 12 billion) (Fig. 2e). However, the greater repurposing of subsidies to a specific agriculture sector also led to reductions in allocative efficiency, in particular in OECD countries (USD 20 billion). In non-OECD countries, the subsidies compensated taxes that are frequently levied on the horticultural sector, which resulted in small increases in allocative efficiency (USD 0.3 billion). The net effect on economic welfare was negative in the OECD under complete repurposing, but mildly positive in non-OECD countries. Repurposing half of the subsidies led to smaller reductions in allocative efficiency, and mitigated most of the net reductions in economic welfare in OECD countries, but also halved health benefits because of less labour-market gains.

**Restructuring of subsidies**. Combining a repurposing of subsidies with a restructuring in which each country provides subsidies in proportion to either its population or GDP, whilst keeping the global amount of subsidy payments fixed, led to increases in the production of fruits and vegetables that were more evenly distributed across regions, with particular large increases in countries without prior subsidies, especially in the population-based subsidy scenario (+4%) (Fig. 2a). Because of the global coverage of the sectoral incentives in this scenario, there were less production-based feedback effects, and overall GHG emissions were reduced similarly or more than in the repurposing scenarios (−0.3% in the GDP scenario, and −0.4% in the population scenario) (Fig. 2b).

The changes in consumption of horticultural products and the associated health impacts were also more equally distributed across regions, in particular in the population-based scenario (Fig. 2c). The increases in fruits and vegetables in previously non-subsidising countries were 12 g/d on average in that scenario (and 5 g/d in the GDP-based scenario), compared to 2 g/d in the repurposing-only scenario. The overall reductions in diet-related mortality were similar in magnitude as the repurposing-only scenario (370,00−379,00 avoided deaths in 2030) (Fig. 2d), but with a more equal distribution of per-capita reductions in mortality (0.2−0.8% in the population-based scenario and 0.1−1.4% in the GDP-based scenario, compared to 0.1−1.5% in the repurposing-only scenario; Supplementary Fig. 9).

Both variants of structural subsidy reform were associated with global increases in economic welfare (USD 1.8−5.5 billion), but regional impacts differed (Fig. 2e). As subsidies were reduced in OECD countries, the reductions in allocative efficiency decreased or turned positive, which led to net economic gains when combined with the gains from an increased labour force. The total level of subsidies stayed similar in non-OECD countries, with impacts similarly positive as in the repurposing-only scenario. However, subsidy payments increased and allocative efficiency decreased in previously non-subsidising countries, which was partly compensated by gains from an increased labour force. The losses could be fully compensated, at least in principle, by transfer payments from other regions as net gains there were twice as large as the net losses in previously non-subsidising countries.

## Discussion

Agricultural subsidies are an important factor influencing production choices. Our analysis suggests that agricultural subsidy reform could make a meaningful contribution to a transition towards healthier and more sustainable food systems, including improvements in population health, environmental pollution and economic welfare. However, trade-offs exist between these impacts when considering different reform options. We found that removing agricultural subsidies could be economically and environmentally beneficial, but it could negatively impact population health. In contrast, redirecting all subsidies to the production of foods with beneficial health and environmental characteristics could improve population health, reduce GHG emissions, but have negative economic impacts. Partial repurposing of subsidies could mitigate economic losses and lead to gains in some scenarios, but it would also be associated with lower health and environmental benefits. Lastly, combining the repurposing of subsidies with a global restructuring of subsidy levels according to GDP or population levels could lead to comparable health benefits as a repurposing-only approach, but with a more equal distribution across regions, similar or greater reductions in GHG emissions, as well as global economic benefits. However, newly subsidising countries would have to be compensated in part to share in those gains.

Our analysis was based on established systems models which are regularly used for policy assessments, and the results were robust with respect to socio-economic, environmental, and health-related uncertainties (Supplementary Tables 13–17). Our study analysed options for agricultural subsidy reform from economic, health, and environmental perspectives in an integrated manner, and our analysis demonstrates the importance of considering the mutual feedbacks across these dimensions. For a repurposing of subsidies, for example, we found that the health-related economic gains from increases in the labour force were often opposed to, and compensated, the losses in allocative efficiency that were associated with greater economic regulation. And although allocative efficiency increased when agricultural subsidies were removed, dietary risks increased as well, and their economic impacts from a loss of labour reduced the savings

from greater economic efficiency. These feedbacks indicate the importance of considering health-related welfare measures when evaluating agricultural policies.

Our analysis is also subject to caveats and raises further points for discussion. First, in many of the reform options, we focused on coupling subsidies to the production characteristics of a specific sector and did not consider differences in agricultural management within that sector, nor impacts beyond GHG emissions. For example, the modelled subsidy schemes would equally reward the production of fruits and vegetables from intensive, monocultural systems with potentially detrimental biodiversity impacts as production based on agro-ecological approaches that are associated with higher local levels of ecosystem services. Ideally, a health and environmentally sensitive subsidy system would incentivise both the broad choice of food commodities that make up healthy and sustainable diets, and the kind of production methods that are most sustainable—both at the local level and when including system-wide and international feedbacks[16]. Further work is also needed to quantify the implications of subsidy reform on other environmental dimensions, including water and pesticide use which some horticultural products do not currently perform well on[13,14,17]. How to couple incentives for transforming the mix of production and consumption that is needed for climate change mitigation with incentives for improving production methods and impacts on other environmental dimensions remains an important avenue for future research.

Second, we focused in our analysis on global, regional, and national-level changes in economic welfare, public health, and GHG emissions, but we did not analyse any sub-national changes that can be relevant for decision making. For example, assessments of how subsidy reforms could change income at the farm level would be especially important for countries where farming is the main source of income for rural households. It is also important to note that in many instances, agricultural subsidies are but one factor influencing production. Others include geographical and climatic conditions, the availability of sufficient resources, the degree of economic integration and globalisation, individual and social preferences, as well as consumer and market demand[18]. Especially the latter represents another important factor that is modifiable by public policies. The level of food system changes required to limit environmental pollution and resource demand in line with environmental limits and stated policy targets will likely require a multicomponent approach with clear incentives for both producers and consumers[4,5].

The extent to which a reform of agricultural subsidies can be implemented will also depend on the political context and will, and on the support by interest groups and the public. As such, the coupling of subsidies analysed here could be politically ambitious, in particular in OECD countries. Over the last decades, subsidy payments there have shifted to decoupled payments as a response to the overproduction of selected commodities[6]. However, these initial approaches to coupling had very different motivations, and recent discussions of agricultural policy reform in the European Union ("Farm to Fork" strategy) and the United Kingdom (new agriculture bill) have stressed the importance of considering the healthiness and environmental sustainability of food production as desirable public goods that are to be supported. A "public money for public goods" approach could make a coupling of subsidies to food commodities that are of high importance for public health and environmental sustainability politically more feasible than past production-centred approaches. Our results suggest that such health and environmentally sensitive approaches to coupling warrant to be seen as important options for a holistic agricultural subsidy reform.

## Methods

**Modelling framework**. We used a coupled modelling framework consisting of economic, environmental and health models to analyse the potential implications of agricultural-subsidy reform. For the economic analysis, we used a computable general equilibrium (CGE) model of the global economy. CGE models combine economic theory and empirical data according to which relative prices of commodities adjust so that supply matches demand across different sectors and regions. They represent the whole economy and include the agriculture sector, as well as industrial and service sectors. Due to their structure, the models allow for the identification of causal effects of policy experiments and other external factors on parameters, such as agricultural output per sector, inter-regional trade, and national consumption.

For this analysis, we used a CGE model with agricultural detail, the Modular Applied General Equilibrium Tool (MAGNET), to estimate the potential impacts that changes in agricultural support measures could have on agricultural trade, production, consumption and economic welfare. One of the main features of MAGNET is the comprehensive representation of land resources as a factor of production, as well as its representation of agricultural policies[11]. Due to its specific focus, MAGNET is regularly employed to inform policy makers and other stakeholders about the economic implications of various policies and other external factors related to agriculture[19–21]. The demand system in MAGNET resolves changes in food demand that are due to changes in the price of a specific food (as represented by own-price elasticities) and due to changes in income (as represented by income elasticities). In MAGNET and other general-equilibrium models, household income is affected by changes in the agri-food system and by changes to other, non-agricultural sectors. A detailed model description is provided in the appendix and by Woltjer and colleagues[11].

As subsidy data we used the OECD's data on producer support estimates (PSE), in particular budgetary transfers as calibrated for analysis with CGE models by the Global Trade Analysis Project (GTAP). PSE components covering market price support, in particular those related to border policies and tariffs, are reflected in GTAP's tariff data and therefore not technically classified as agricultural subsidies in the model. In line with the GTAP implementation, our representation of budgetary transfers differentiates between the type of payments and the degree of coupling, i.e. whether they are output payments, intermediate input payments, land-based payments, capital-based payments, and labour-based payments, as well as what their degree of coupling is, which is a measure of how far payments are tied to the production of a specific agricultural good.

In the MAGNET model aggregation, the global economy is subdivided into 28 countries and regions (Table S2) and 34 sectors (Table S3). For analysing the health and environmental impacts of agriculture-subsidy reforms, we downscaled the economic feedbacks obtained from the MAGNET model to the country level. We disaggregated the production data using production estimates of the Food and Agriculture Organization of the United Nations (FAO), and we disaggregated the consumption data using FAO estimates of the amount of food available for human consumption, adjusted for food waste at the household level[22,23]. Our framework traces the chain of food from primary production through intermediate sectors to final consumption. For example, vegetables can be purchased raw from the primary sectors, as canned vegetables from the processed food sector, or as part of a prepared meal from the restaurant sector.

We used a global risk-disease model with country-level detail to estimate the impacts that dietary changes related to agricultural-subsidy reform could have on disease mortality. The model uses a comparative risk assessment method which relates changes in risk factors, such as reductions in the consumption of fruits and vegetables, to changes in cause-specific mortality, such as cancer and coronary heart disease[24]. The same concept forms the basis of the Global Burden of Disease project that tracks the impacts of different risk factors on mortality and morbidity in different regions and globally[1].

The comparative risk-assessment model used here included eight diet and weight-related risk factors and five disease endpoints. The risk factors were high consumption of red meat, low consumption of fruits, vegetables, nuts, legumes, and fish, as well as being underweight, overweight, and obese, the latter of which are related to changes in energy intake. The disease endpoints were coronary heart disease (CHD), stroke, type-2 diabetes mellitus (T2DM), cancer (in aggregate and as colon and rectum cancers), and respiratory disease. We adopted relative risk estimates that relate changes in risk factors to changes in disease mortality from meta-analyses of prospective cohort studies to minimise bias from individual studies[15,25–31]. A detailed model description is provided in the appendix and by Springmann and colleagues[12].

For representing the feedback between health and economic impacts, we used the estimated number of diet-related deaths amongst the total population and amongst those of working age to adjust the population and workforce parameters in the economic model and re-ran it, which had implications for economic output and welfare measures.

For analysing the environmental impacts of agricultural subsidy reform, we paired the down-scaled changes in production and consumption derived from the economic model with a set of environmental footprints[5]. We focus on changes in GHG emissions in the main text as those most directly related to dietary changes and are relatively less modifiable by changes in farm-level management[5]. Agricultural GHG emissions include methane and nitrous oxide emissions, but they exclude carbon-dioxide emissions which, following the methodology of the International Panel on Climate Change, are allocated to the energy or other sectors. The footprints for animal source foods include the indirect impacts related to feed production and the direct impacts related to methane emissions. The projections of environmental footprints to the year 2030 included the adoption of technologies and improvements in management practices in line with socio-economic trajectories[5].

**Model scenarios.** Based on these data, we constructed three overall scenarios of agricultural subsidy reform:

(1) Removal of subsidy payments (RMV):

All subsidy payments are removed in this scenario to analyse the impacts of the existing subsidy scheme as a counterfactual.

(2) Repurposing of subsidy payments (S50, S100):

Different shares of the overall subsidy budget are redirected to low-emitting and nutrition-sensitive food commodities (vegetables, fruits, legumes, and nuts) in a budget neutral manner. The shares of repurposing range from 50% where half of subsidies are redirected and half are preserved, to 100% where all subsidies are redirected.

(3) Repurposing of subsidy payments combined with redirecting them globally (GDP, POP):

In scenario 2, we assumed constant overall agricultural subsidy budgets in all countries that have a subsidy scheme. However, subsidy payments are very unequally distributed across countries and regions. In the third set of scenarios, we modelled a more equal distribution of subsides globally. For that purpose, we fixed subsidy budgets globally but implemented them in all countries according to either their GDP (GDP scenario) or their population shares (POP scenario). This implies that now also countries that did not subsidise agriculture initially will implement some support payments. In a given country, the payments are used as subsidies for producing nutrition-sensitive and low-emitting food commodities (vegetables, fruits, legumes, and nuts).

We analysed the production, consumption, environmental, health, and economic welfare implications of each of the scenarios for a target year of 2030. The baseline trajectory to 2030 takes into account projections of a middle-of-the-road growth path of population and real GDP[32], including labour growth, and projections of biophysical yield developments of crops and pastures caused by climate and area changes[33]. The representation of agricultural policies and data on agricultural subsidy payments were based on updated data for the year 2017, in line with agreed-to policies and the latest release of PSE estimates by the OECD. For a structured uncertainty analysis, we used different socio-economic development pathways that included more optimistic ones with greater economic and lower population growth, and more pessimistic ones with lower economic and higher population growth[32]. We did not include parameter uncertainty within these pathways.

**Reporting summary.** Further information on research design is available in the Nature Research Reporting Summary linked to this article.

## Data availability

The results data generated in this study have been deposited in the Oxford University Research Archive (ORA) available at https://doi.org/10.5287/bodleian:QmzkJxaYZ.

## Code availability

The code of the health and agriculture-economic models is described in detail in the Supplementary Information and the references cited therein. Because the MAGNET model is licensed, we can only make the full code available upon request.

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

## Acknowledgements

We gratefully acknowledge helpful comments from Patrick Webb and Matin Qaim. This research was funded by the Global Panel on Agriculture and Food Systems for Nutrition (GLOPAN) (MS, FF) and the Wellcome Trust, Our Planet Our Health (Livestock, Environment and People (LEAP)), award number 205212/Z/16/Z (MS).

## Author contributions

Conceptualization, F.F. and M.S.; methodology, F.F. and M.S.; investigation, F.F. and M.S.; writing—original draft, M.S.; writing—review & editing, F.F. and M.S.

## Competing interests

The authors declare no competing interests.
