## [Peer Review File · Nature Communications]

Dear reviewers,

Thank you for your comments and suggestions on our manuscript. We have substantially revised the manuscript in response. A major aspect of the revision concerns the data on subsidies. In coordination with colleagues from the Global Trade Analysis Project, we updated the specification of subsidies in the agriculture-economic model. This allowed us to use more up-to-date data and data for more regions, including India. In addition, we clarified the model description and the presentation of results, and we expanded the Discussion section and the Supplementary Information. The following point-by-point response details the changes made in response to each of the reviewers' comments.

Response to reviewers' comments:

Referee #1:

Comment 1-0:

General Thank you for the opportunity to review this interesting article on options for reforming agricultural subsidies from health, environmental, and economic perspectives. Aligning agricultural subsidies with societal goals including health and the environment is an important area of research, and the authors have undertaken the related quantitative analyses – presenting various options for reforming agricultural subsidies. I'm not in a position to be able to review the modelling undertaken, but have provided some suggestions in relation to other aspects of the article.

Reply to Comment 1-0:

Thank you for your helpful comments and suggestions. Please find a point-by-point response below.

Comment 1-1:

The authors say that they find several reform options that could lead to environment and health improvements without reductions in economic welfare and suggest that a more equal distribution of subsidy payments globally would help achieve this. However, from what I can tell, this equal distribution of subsidy payments would be in fact a redistribution, and would benefit some countries and economies at the expense of others. Thus, the description of this being achieved 'without reductions in economic welfare' requires more explanation.

Reply to Comment 1-1:

You are right in pointing out that not all regions might experience economic gains when subsidy payments would be distributed more equally. Our results suggest that countries who currently subsidise agriculture, whether in the OECD or not, would gain under a restructuring, but countries that currently do not provide subsidies could experience economic losses if they started subsidising. Allowing countries to subsidise agriculture to a level that is proportional to their GDP minimised those losses.

We'd like to note that the reform scenarios that involve a restructuring do not include a redistribution, but a change in each country's allowed limit for providing subsidies. We mentioned the possibility of redistribution to compensate the countries that would experience economic losses when newly subsidising, because global economic gains exceeded losses, which makes such an option possible. The specific part in the abstract that you are referring to in the

comment is written from this global perspective that also holds for currently subsidising countries. Following the comment, we inserted “on a global scale” to clarify this.

In addition, we more fully explain the option for compensating payments in the summary paragraph of the Discussion section as follows:

“Lastly, combining the repurposing of subsidies with a global restructuring of subsidy levels according to GDP or population levels could lead to comparable health benefits as a repurposing-only approach, but with a more equal distribution across regions, similar or greater reductions in GHG emissions, as well as global economic benefits. However, newly subsidising countries would have to be compensated in part to share in those gains.”

Comment 1-2:

As the authors describe, agricultural subsidies are an important factor influencing production choices. The describe how their analysis suggests that agricultural subsidy reform could make a meaningful contribution to a transition towards healthier and more sustainable food systems, including improvements in population health, environmental pollution and economic welfare. It would be helpful if the authors could position agricultural subsidies within the wider range of factors influencing production choices. This would help the reader understand the context in which ag subsidies could make the meaningful contributions described – and any factors within the wider system that may prevent that from happening. I say this, mindful of the need for more of a systems perspective to the description of subsidies – which are simple one part of a wider ag-food-economic-enviro system.

Reply to Comment 1-2:

Thank you for the suggestion. We inserted a new paragraph in the Discussion section to provide additional context:

“It is also important to note that in many instances, agricultural subsidies are but one factor influencing production. Others include geographical and climatic conditions, the availability of sufficient resources, the degree of economic integration and globalisation, individual and social preferences, as well as consumer and market demand ¹. Especially the latter represents another important factor that is modifiable by public policies. The level of food system changes required to limit environmental pollution and resource demand in line with environmental limits and stated policy targets will likely require a multicomponent approach with clear incentives for both producers and consumers ^{2,3}.”

Comment 1-3:

The authors discuss trade-offs between the different agricultural subsidy reform options – but what about impact that the reform options may have on wider factors influencing production choices? As this is all part of a complex system, changes to agricultural subsidies may result in unintended or unforeseen consequences for other parts of the system – and may not thus have the effect modelled here. This needs to be considered in the paper, even if not modelled.

Reply to Comment 1-3:

We agree that interventions in the agricultural sector in one country can have feedbacks on production and consumption in other countries, and on other sectors of the economy. For example, increasing production in one part of the economy by increasing subsidies, e.g. in agriculture, can lead to a reduction in production in other parts of the economy, e.g. industry, as resource endowments are fixed. The modelling framework we used represents such “general equilibrium” effects. We provide a detailed model description in the Supplementary Information, and we summarise the model structure in the Methods section as follows:

“For the economic analysis, we used a computable general equilibrium (CGE) model of the global economy. CGE models combine economic theory and empirical data according to which relative prices of commodities adjust so that supply matches demand across different sectors and regions. They represent the whole economy and include the agriculture sector, as well as industrial and service sectors. Due to their structure, the models allow for the identification of causal effects of policy experiments and other external factors on parameters, such as economic output per sector, inter-regional trade, and national consumption.”

Please let us know if those were the kind of feedback that you had in mind. We’d be happy to provide additional explanations if needed.

Comment 1-4:

Policies are not just made based on evidence – the article should consider the wider political factors likely to impact the uptake of the different options described.

Reply to Comment 1-4:

We agree, and following the comment, we extended the paragraph in the Discussion section that covers the wider political factors. It now reads:

“The extent to which a reform of agricultural subsidies can be implemented will also depend on the political context and will, and on the support by interest groups and the public. As such, the conditioning of subsidies analysed here could be politically ambitious, in particular in OECD countries. Over the last decades, subsidy payments there have shifted to decoupled payments as

a response to the overproduction of selected commodities ⁴. However, these initial approaches to coupling had very different motivations, and recent discussions of agricultural policy reform in the European Union (“Farm to Fork” strategy) and the United Kingdom (new agriculture bill) have stressed the importance of considering the healthiness and environmental sustainability of food production as desirable public goods that are to be supported. A “public money for public goods” approach could make a conditioning of subsidies to food commodities that are of high importance for public health and environmental sustainability politically more feasible than past production-centred approaches. Our results suggest that such health and environmentally sensitive approaches to “coupling” warrant to be seen as important options for a holistic agricultural subsidy reform.”

Comment 1-5:

First sentence of the Abstract – I think the words ‘are a’ are missing after ‘therefore’.

Reply to Comment 1-5:

Thank you. We corrected the typo.

References

1. Food, U. N. & Organization (FAO)(Rome), A. *FAO/WFP Joint Guidelines for Crop and Food Security Assessment Missions (CFSAMs)*. (FAO, 2009).
2. Willett, W. *et al.* Food in the Anthropocene: the EAT–Lancet Commission on healthy diets from sustainable food systems. *The Lancet* **393**, 447–492 (2019).
3. Springmann, M. *et al.* Options for keeping the food system within environmental limits. *Nature* **562**, 519–525 (2018).
4. OECD. *Agricultural Policy Monitoring and Evaluation 2018*. (2018). doi:10.1787/agr_pol-2018-en.

Referee #2:

Comment 2-0:

A. Summary of the key results

Removing or changing agricultural subsidies may have important effects on diets. Simulations suggest that especially subsidies on fruits, vegetables and nuts have large effects on their production and consumption and therefore on health effects of diets. Abandonment of subsidies may therefore have negative health effects. Redirecting subsidies to fruits, vegetables and nuts gives positive health effects and climate benefits, but seems to have negative economic impacts. Both effects are straightforward. Equalising subsidies over countries based on population or GDP (keeping the same budget) increase global welfare. However, the countries who start with subsidies have a welfare loss. These welfare effects of subsidies (where effects on health are not included, as far as I understand) are a standard result of CGE analysis.

In order to do the analysis a health model and a general equilibrium economic model have been coupled and for environmental effects LCA results are used.

B. Originality and significance: if not novel, please include reference

As far as I know no one has done an exercise with respect to the relation between agricultural subsidies, health and greenhouse gasses. No one did couple a CGE model with health model and relating this with labour supply, as far as I know. However, I think that the general results are very straightforward. Subsidies imply a loss in equivalent variation (economic loss), but if they are targeted towards products that are better for health and climate, they give benefits on those terrains. This is not too surprising.

C. Data & methodology: validity of approach, quality of data, quality of presentation

A lot of effort is put in the exercise. However, in the main text only provides many numbers without an intuitive explanation of the mechanisms behind it and numbers which are highly uncertain.

D. Appropriate use of statistics and treatment of uncertainties

Uncertainties with respect to scenarios are included. As far as I see not the uncertainties with respect to parameters, which will be large.

F. Suggested improvements: experiments, data for possible revision

G. References: appropriate credit to previous work?

Yes

H. Clarity and context: lucidity of abstract/summary, appropriateness of abstract, introduction and conclusions

Abstract is clear. Introduction is very general. Conclusions are included in discussion.

Reply to Comment 2-0:

Thank you for your helpful comments and suggestions. In our revision, we have aimed to provide more detailed explanation of our methods and results. Please find a point-by-point response below.

Comment 2-1

Scenario descriptions could be more precise. For example, it could be explained in the main text that in scenario S100 all subsidies go into fruits, nuts and vegetables (as shown in table S9), (while in the benchmark about 20% of subsidies is allocated to horticultural products), instead of the abstract formulation “Repurposing of 25-100% of agricultural subsidies to support food commodities with beneficial health and environmental characteristics”.

Reply to Comment 2-1:

Following the comment, we clarified what we mean with repurposing at the first instance. The results section on repurposing now reads as follows:

“Using agricultural subsidies (i.e. repurposing from previous ways of allocating subsidies) to support the production of foods with beneficial health and environmental characteristics led to ...”

Comment 2-2:

It would also be useful to explain better how consumption is determined: what is the share of indirect consumption of agricultural products in total consumption? Also a list of demand elasticities would be useful, at least for the three main product categories. And added to that it would be useful to know the size of the subsidies as percentage of value added in order to get an idea of the price effect. For me it is surprising that the effect of subsidy removal from meat products is so much smaller than subsidy removal from horticulture, while the size of the sector is not that different and demand elasticities are probably higher in the meat related sectors.

Reply to Comment 2-2:

In our analysis, food consumption is determined by tracing primary production via intermediate sectors to the final consumer. For example, fruits and vegetables can be purchased directly from the primary production sector (denoted as “f&v” in our model aggregation), or for example as canned vegetables from the processed food sector (denoted as “ofd” for other processed food), or as part of a prepared meal in the restaurant sector (“sevcs”). Following the comment, we added this explanation to our Methods description and to the Supplementary Information.

As requested, we also added a table of the compensated demand elasticities we used (Table S1), and of price changes (Table S12). The reason that the effect of subsidy removal on meat is smaller than for horticulture is due to the fact that horticultural consumption is much larger in absolute (grams per day) terms than meat consumption, whilst the percentage reductions for both food categories are comparable. For example, global vegetable consumption is around 400

g/d while beef consumption is around 30 g/d. Following the comment, we added a table with baseline consumption values (in grams per day) to the Supplementary Information in support of this explanation (Table S5).

Comment 2-3:

It is not clearly described how it is handled that not all production is consumed in the standard GTAP input-output tables; is all consumption allocated to the households by redefining the input-output tables? How are corrections made to make the percentage changes of consumption exactly the same as percentage changes of production?

Reply to Comment 2-3:

In general, the input-output tables and the CGE model structure assure that everything that is produced is also consumed in one way or another. However, not all agricultural products are consumed by the private household directly but could be further processed for non-food purposes like bioenergy or biochemical uses, which are also part of the model. In addition to representing intermediate processes and non-food uses, we also accounted for the amount of food that is wasted at the household level by using data from the FAO. We describe the latter in more detail in the Supplementary Information, and we added more detail on how intermediate sectors are represented in response to the previous comment.

Comment 2-4:

It is not clear how substitution of food is handled. In the CDE consumption function the cross price elasticities are low and the result of own price elasticities and income elasticities, while in practice an demand increase of horticultural products will be at the cost of some other food commodities. Information on percentage price changes of the different commodities as intermediate variable explaining the effect.

Reply to Comment 2-4:

Thank you for this comment. The substitution of food is a result of general equilibrium effects in our model. For instance, when subsidies are redirected to incentivise greater production of horticultural products, then less land and other production factors are available for other crops. This drives up the prices of those crops, which lowers their consumption. Following the comment, we expanded our explanation of such effects in the Methods section, and we appended an overview of changes in consumer prices in the Supplementary Information (Table S12). The revised Methods section now reads in this regard:

“The demand system in MAGNET resolves changes in food demand that are due to changes in the price of a specific food (as represented by own-price elasticities), in the price of other foods (as

represented by cross-price elasticities), and due to changes in income (as represented by income elasticities). In MAGNET and other general-equilibrium models, household income is affected by changes in the agri-food system and by changes to other, non-agricultural sectors.”

Comment 2-5:

In the table below (made from the additional material, where explanation of the meaning of all abbreviations could be more complete) I see that the negative health effect of eating red meat is about half of the beneficial effect of eating more vegetables. [...] In the table below one explanation may be that the amount of fruits and vegetables eaten is very high: 549 gram per day. Relatively simple formulas must have been used that relate changes in consumption with deaths as a result of diet changes. It would be useful if they would be visible. I have some problem in interpreting the relative risk factors in table S5 in this context. I think it can be directly translated into avoided death per extra gram consumption. For lay people on this topic it would be informative to have these numbers.

Reply to Comment 2-5:

We used a comparative risk assessment for estimating the health impacts of changes in dietary risk factors. The assessment uses epidemiological data, including relative risk values that relate changes in risk factors (e.g. increased consumption of vegetables) to changes in the risk of dying from specific diseases. In addition, the calculations use disease-specific mortality rates and population numbers, each differentiated by age group and country. We provided the formula with which the change in diet-related disease burden is calculated in the Supplementary Information (see the section “Health analysis”). Following your comment, we also prepared an illustrative example that we discuss below.

Relative risk values are derived from epidemiological cohort studies (and meta-analyses of those) and describe the general dose-response relationship between a risk factor and a health outcome. As Table S7 (formaly S5) in the SI shows, a 100 g/d increase in red meat consumption is associated with an increase in the risk of dying from coronary heart disease (CHD) of 15% on average (RR=1.15, 95% confidence interval: 1.08-1.23), whereas a 100 g/d increase in vegetable consumption has been associated with a reduction in the risk of dying from CHD of 16% (RR=0.84, 95% confidence interval: 0.80-0.88).

What matters for calculating the diet-related disease burden is the change in risk exposure, i.e. the change (in grams per day) in vegetable and red-meat consumption (in line with the equation for calculating population impact fractions, PIFs, provided in the SI). In the EU, for example, vegetable consumption increased by 38 g/d under complete repurposing (see the change (“chg”) of the waste-adjusted consumption parameter (“g/d_w”) in the S100 scenario), whilst red meat consumption (which includes beef, lamb, pork) decreased by 3.4 g/d.

Multiplying the population impact fractions by mortality rates and population numbers (as described in the SI) indicates that the increase in vegetable consumption would be associated with 41,000 avoided deaths from CHD, and the reductions in red meat would be associated with 3,600 avoided deaths from CHD. Thus, the order of magnitude difference in the changes in consumption was roughly carried through to the final health estimates.

As described in the Supplementary Information, the exact calculations also account for other disease associations (Table S7), co-exposure to multiple risk factors (as death has to be ascribed to one specific risk factor in the assessment; see PIF equation), age-effects that attenuate risks at older ages, age-effects that differentiate mortality rates, as well as non-linear dose-response functions that reduce the impacts of an increase, e.g. in vegetable consumption, when baseline consumption is already relatively high.

Comment 2-6:

In order to get the conclusions, you don't need much of the model, only the basic assumptions used. These are the current subsidies, an intuition on the order of magnitude of demand elasticities, and the supply elasticities. This could be more clearly formulated.

Reply to Comment 2-6:

We agree that examining a model's key parameters can provide a basic intuition on general trends and the orders of magnitude of interactions when all other things are held constant. This is especially the case in partial equilibrium analysis. However, in the type of general equilibrium analysis we did, everything depends on each other. For example, supply elasticities are not simply constant parameters but are complicated functions of the model's parameter space and functional forms. Similarly, the modelled health effects depend not only on relative risks and changes in exposure, but on a range of parameters, including the interactions handed down from the general equilibrium analysis. Following your comment, we clarified the description of our model framework and provided additional detail on the feedbacks and interactions represented by it. Please see in particular our replies to the two previous comments.

Comment 2-7:

The effects of the subsidies on consumption and health are relatively uncertain. This could be stated more clearly.

Reply to Comment 2-7:

We followed established standards of reporting uncertainty related to health assessments. Our health analysis accounted for epidemiological uncertainty (see Table S7 in the Supplementary

Information), and we reported the results as mean impacts including 95% confidence intervals. We explain this in the Supplementary Information as follows:

“For the different diet scenarios, we calculated uncertainty intervals associated with changes in mortality based on standard methods of error propagation and the confidence intervals of the relative risk parameters.”

To analyse the robustness of our results, including parameter-specific uncertainties, we also included a structured sensitivity analysis with different socio-economic pathways. We mention that in the Discussion section as follows:

“Our analysis was based on established systems models which are regularly used for policy assessments, and the results were robust with respect to socio-economic, environmental, and health-related uncertainties (Tables S13-S17).”

Comment 2-8:

It is not clear to me why subsidies have such enormous effects on health effects of vegetables and fruits, and why abolition of meat subsidies has relatively small effects. The conclusion that abolishment of agricultural subsidies results in a reduction in health because of less consumption of horticultural products seems not very plausible to me. Global consumption of on average 600 grams of fruits and vegetables per person per day seems extremely high.

Reply to Comment 2-8:

Please see our response to Comment 2-5.

Regarding the consumption estimates: as explained in the Methods and Supplementary Information, we estimated baseline food consumption by adopting estimates of food availability from the FAO’s food balance sheets, and adjusting those for the amount of food wasted at the point of consumption^{1,2}. Food balance sheets report on the amount of food that is available for human consumption². They reflect the quantities reaching the consumer, but do not include waste from both edible and inedible parts of the food commodity occurring in the household, which is why we adjusted those estimates for waste and processing.

An alternative would have been to rely on a set of consumption estimates that has been based on a variety of data sources, including dietary surveys, household budget and expenditure surveys, and food availability data^{3,4}. However, neither the exact combination of these data sources, nor the estimation model used to derive the data have been made publicly available. For some individual countries, using dietary surveys would also have been an alternative. However,

underreporting is a persistent problem in dietary survey^{5,6}, and regional differences in survey methods would have meant that our results would not be comparable across countries.

In contrast to dietary surveys, waste-adjusted food-availability estimates indicate levels of energy intake per region that reflect differences in the prevalence of overweight and obesity across regions⁷, which provides an additional test for general consistency.

References

1. Gustavsson, J., Cederberg, C., Sonesson, U., Van Otterdijk, R. & Meybeck, A. *Global food losses and food waste: extent, causes and prevention*. (2011).
2. Food and Agriculture Organization of the United Nations. *Food balance sheets: a handbook*. (2001).
3. Gobbo, L. C. Del *et al.* Assessing global dietary habits: a comparison of national estimates from the FAO and the Global Dietary Database. *The American Journal of Clinical Nutrition* **101**, 1038–1046 (2015).
4. Micha, R. *et al.* Global, regional and national consumption of major food groups in 1990 and 2010: a systematic analysis including 266 country-specific nutrition surveys worldwide. *BMJ Open* **5**, e008705 (2015).
5. Freedman, L. S. *et al.* Pooled results from 5 validation studies of dietary self-report instruments using recovery biomarkers for energy and protein intake. *American journal of epidemiology* **180**, 172–188 (2014).
6. Rennie, K. L., Coward, A. & Jebb, S. A. Estimating under-reporting of energy intake in dietary surveys using an individualised method. *British Journal of Nutrition* **97**, 1169–1176 (2007).
7. NCD Risk Factor Collaboration (NCD-RisC). Trends in adult body-mass index in 200 countries from 1975 to 2014: a pooled analysis of 1698 population-based measurement studies with 19.2 million participants. *The Lancet* **387**, 1377–1396 (2016).

Referee #3:**Comment 3-0:**

This is a really interesting paper that tries to answer an important question on the impact of restructuring agricultural subsidies on food prices, diets, human health, and the environment. The paper is novel because it looks at the environmental and health impact as well as the impact on human welfare—not only on a global scale but also across different parts of the world.

I have three suggestions for authors to consider while revising the paper. I discuss them below.

Reply to Comment 3-0:

Thank you for your helpful comments and suggestions. Please find a point-by-point response below.

Comment 3-1:

Underestimation of agricultural subsidies in the non-OECD countries

The subsidy data provided by the OECD for the year 2017 appears inaccurate, especially, for the non-OECD countries. According to the paper, non-OECD countries spend around \$ 59 bn on subsidies (28% of \$ 211 billion)—with China accounting for the largest share of it. India is not mentioned in the list, but my estimates suggest that India alone spends more than \$ 30 billion per year on subsidizing chemical fertilizers (\$ 11 billion), electricity for irrigation (\$ 11.4 billion in 2013-14), and price support for food grains (\$ 4.5 billion). Since 2019, the Government of India has started another large crop-neutral subsidy program to transfer around \$10-12 billion/year to farmers as income support. With this new addition, India's total agricultural subsidy will exceed \$ 40 billion/year—close to what China pays its farmers. Therefore, as per the government data, agricultural subsidies are high in India (>10% of agricultural GDP), but it is likely that the analysis in this paper classifies India into one of the non-subsidizing countries.

Even without this misclassification, if the model is based on an under-estimation of different types of agricultural subsidies and their distribution across the countries in the world, the simulation results on how a partial or complete removal of subsidies will affect crop outputs in different parts of the world will also be off. This misestimation will, in turn, influence model results on food trade dynamics and food prices—globally and in 'subsidizing' and non-subsidizing' countries.

Reply to Comment 3-1:

Our initial analysis included subsidy data for the year 2017 for those countries that subsidised agriculture in the base year of 2014. This was done for computational reasons. We do agree that this underestimated the potential impacts of subsidy reform especially for

countries for which subsidy estimates had not been available in the base year, including India.

Following the comment, we worked together with staff from the Global Trade Analysis Project (GTAP) to include subsidy data also for countries from OECD's PSE database which had not been available in the previous version of the paper. As a consequence, we also changed the model aggregation to include these new countries as individual regions. Our revised manuscript now contains country-specific estimates, both of current subsidies and the impacts of subsidy reform, for India, Colombia, Philippines, and Viet Nam. Including new data, especially for a large country such as India, affected the impacts for the group of non-OECD countries, but the general trends remained similar.

Comment 3-2:

The income effect of repurposing or restructuring subsidies on the food producers. It is not clear to me if the model accounts for the income effect of the change in the subsidy regime on food producers. Even if the changes proposed/simulated in the paper are revenue neutral, they may have significant effects on the incomes of farmers. In OECD countries, where only a small percentage of families are into farming, this analysis is not required. However, in poor countries where farming is the main source of employment and income for a majority of rural households, the income effect of changing subsidies would matter—separately from the effect on consumer prices of food.

I am not a modeler. So, it is possible that I missed this analysis while reading the paper and the supplementary material. I would suggest that the authors account for and clearly describe how the income effect on producers plays out. These income effects are also important to understand because they will affect the political viability of the types of changes simulated in the paper. A number of developing countries are trying to switch from price-distortionary subsidies to crop-neutral direct cash transfers or income transfers to farmers, but these changes are facing stern opposition from farmers who fear income loss or an increase in uncertainty from such changes.

Reply to Comment 3-2:

We agree that analysing how farm income would change in response to subsidy reform is an interesting and important avenue for research. Unfortunately, our model framework is not best suited to undertake such an analysis, and dedicated farm-level models would be needed to study changes in farm income. What we were interested in were national-level changes in economic welfare, public health, and greenhouse gas emissions. Whilst our model framework resolves price-driven changes in supply and demand for different sectors, we consider any farm-level analysis beyond the scope of the present study. However,

following your comment, we now mention farm-level analysis as an important avenue for further research in the Discussion section:

“[W]e focused in our analysis on national-level changes in economic welfare, public health, and greenhouse gas emissions, but we did not analyse any sub-national changes that can be relevant for decision making. For example, assessments of how subsidy reforms could change income at the farm level would be especially important for countries where farming is the main source of income for rural households.”

Comment 3-3:

The treatment of environmental impacts of food production in the model
As subsidies change in the model simulations, production levels of different types of foods change, but the emission footprints remain fixed at the baseline level. For projecting future GHG emissions, the model incorporates the mitigation potential of changes in agricultural technologies and practices by comparing the abatement cost curves and the social value of carbon. It's not clear to me if the future abatement costs in the model change with the changes in the subsidy regime.

A big reason why governments are trying to replace price distortionary subsidies (like the subsidy on the price of Urea in all countries of South Asia) with direct cash transfer is that the resulting change in incentives will lead to the more efficient application of Urea. More crop will be produced with the use of less Urea. Similarly, it is expected that the change in water (or energy prices for pumping groundwater) will lead to the production of more crops per drop due to the faster adoption of water saving technologies and practices.

Does the model capture this change in input use efficiency, and hence the emission footprints, of food production with the restructuring and repurposing of subsidies? I would request the authors to consider it if it has not already been done in the paper. They should also highlight in the text how the change in subsidies will change environmental footprints or pollution load per ton of food produced.

Reply to Comment 3-3:

Our analysis was conducted at the level of food groups, and we incorporated changes in the environmental footprints of food groups in line with economically driven changes in management practices. Our focus was on the changes in the mix of foods that are produced and consumed, keeping the relative resource requirements for each food category in line with projections along socio-economic trajectories.

For context, the projected improvements in management practices resulted in a reduction in food-related greenhouse gas emissions of 6% on average in the middle-of-the-road socio-

economic trajectory, ranging from 3-9% for countries grouped by income (i.e., from high to low-income countries). We now list these values in the Supplementary Information. Our view is that such changes are part of the expected changes in management practices irrespective of changes in the way agricultural subsidies are governed.

The specific changes in Urea application you mention are surely interesting, but we would argue that analysing subsidy regimes that are tied to specific changes in management practices are beyond the scope of the current analysis and would require a dedicated analysis of management-level options for improving agriculture efficiency. Following your comment, we mention this line of work as an important aspect for future research in the Discussion section:

“[I]n many of the reform options, we focused on conditioning subsidies to the production characteristics of a specific sector and did not consider differences in agricultural management within that sector, nor impacts beyond GHG emissions. For example, the modelled subsidy schemes would equally reward the production of fruits and vegetables from intensive, monocultural systems with potentially detrimental biodiversity impacts as production based on agro-ecological approaches that are associated with higher levels of ecosystem services. Ideally, a health and environmentally sensitive subsidy system would incentivise both the broad choice of food commodities that make up healthy and sustainable diets, and the kind of production methods that are most sustainable – both at the local level and when including system-wide and international feedbacks¹.”

References

1. Balmford, A. *et al.* The environmental costs and benefits of high-yield farming. *Nature Sustainability* **1**, 477–485 (2018).

Referee #4:**Comment 4-0:**

The authors apply an optimization model that shows them that a shift of agricultural subsidies towards more environmentally friendly and healthy products would lead to healthier diets and a better environmental situation. I find it even more interesting that the situation would also improve if subsidies would be shifted to other countries. While I generally believe that there is merit in the manuscript, I see some points that need clarification.

Reply to Comment 4-0:

Thank you for your helpful comments and suggestions. Please find a point-by-point response below.

Comment 4-1:

It has been shown before that a different portfolio of agriculture helps to avoid a lot of emissions (eg. A. von Ow, T. Waldvogel, T. Nemecek 2020 in the Journal of Cleaner Production). Such knowledge should be used for the study design.

Reply to Comment 4-1:

Thank you for the suggestion. Our focus was on how changes in subsidy regimes would affect changes in the mix of foods that are produced and consumed, keeping the relative resource requirements (and the associated management practices) for each food category in line with projections along socio-economic trajectories. Analysing subsidy regimes that are tied to specific changes in management practices are surely interesting and relevant, but we would argue that they are beyond the scope of the current analysis. At the same time, we acknowledge the importance of considering these aspects in the Discussion section as follows:

“Our analysis is also subject to caveats and raises further points for discussion. First, in many of the reform options, we focused on conditioning subsidies to the production characteristics of a specific sector and did not consider differences in agricultural management within that sector, nor impacts beyond GHG emissions. For example, the modelled subsidy schemes would equally reward the production of fruits and vegetables from intensive, monocultural systems with potentially detrimental biodiversity impacts as production based on agro-ecological approaches that are associated with higher levels of ecosystem services. Ideally, a health and environmentally sensitive subsidy system would incentivise both the broad choice of food commodities that make up healthy and sustainable diets, and the kind of production methods that are most sustainable – both at the local level and when including system-wide and international feedbacks ¹.”

We would also like to note that although the economic-agriculture model we used features some level of optimisation, that aspect is purely limited to the representation of the economy, where it is used to ensure that supply matches demand. This process enables the representation of economic feedbacks. In comparison to the reference cited, we were not interested in finding an environmentally optimal configuration of national production practices. Instead, we wanted to assess the potential impacts of reforming subsidies along the dimension of how they are distributed across countries and coupled to the production of specific food groups.

Comment 4-2:

The idea that vegetables, fruits and nuts are the most wonderful crops comes about as an ex-ante assumption. In fact, fruits are usually much more pesticide intensive than cereals. More LCA knowledge should flow in the treatments of the different crops.

Reply to Comment 4-2:

Our focus on coupling subsidies to the production of fruits and vegetables was motivated by considerations of food system transformation as mentioned in the Introduction:

“Model-based analyses suggest that in addition to technological innovation and changes in farming practices, also large-scale dietary changes and concomitant changes in agricultural production will be needed to achieve healthy diets for a growing population, whilst staying within the environmental limits of the food systems². For example, instead of additional global increases in the production of staple crops, animal-source foods, and sugar crops – estimated at 40-80% between 2010 and 2050 – a food system underpinning healthy and sustainable diets would require shifts from those food groups to foods that are both healthy and lower in environmental resource use and pollution, such as fruits, vegetables, legumes, and nuts and seeds.”

We do not claim or assume that fruits and vegetables are the most wonderful crops, but wanted to adopt a food system perspective at the level of food groups as an important level of decision making. At that level, many horticultural products are associated with clear public-health benefits, whilst having relatively low environmental impacts, especially when considering greenhouse gas emissions in comparison to the current mix of production. We explain this approach in the Introduction as follows:

“For the conditioning of subsidies, we adopted a food-group approach and, in line with projections of the required food-system transformation for healthy and sustainable diets, redirected different proportions of subsidies to the production of horticultural commodities (fruits, vegetables, legumes, nuts) that have been associated beneficial health and

environmental characteristics. Approaches that focus on conditioning subsidies explicitly to the actual health and environmental characteristics of food commodities are, despite some differences, to a large degree analogous to our more sectoral food-group approach. For example, life-cycle analyses indicate that although production practices play a role in the environmental impacts of the food system, the impacts of what type of food is grown far outweighs how it is grown – a trend that is especially pronounced when comparing animal source foods with plant-based ones, and when comparing different foods within the same region^{3,4}. A similar trend has been identified for dietary risks and the associated health impacts: non-starchy plant-based foods such as fruits, vegetables, legumes, and nuts have been clearly associated with reduced risks for various diet-related diseases, while other foods have either been associated with increased risk (red and processed meat) or are seen as relatively risk neutral (poultry and dairy) compared to baseline diets^{5,6}. Here we focus on these general health and environmental characteristics, not excluding that additional differentiation might sometimes be appropriate.”

In line with the comment, we agree that the environmental impacts, especially of fruits and vegetables, can be moderate to high for specific indicators such as water and pesticide use. Following the comment, we expanded the Discussion section by adding our focus on greenhouse gas emissions as a caveat and mentioning the importance to consider other environmental impacts in future studies:

“Further work is also needed to quantify the implications of subsidy reform on other environmental dimensions, including water and pesticide use which some horticultural products do not currently perform well on^{3,4,7}. How to couple incentives for transforming the mix of production and consumption that is needed for climate change mitigation with incentives for improving production methods and impacts on other environmental dimensions remains an important avenue for future research.”

Comment 4-3:

Given this last remark, I wonder if greenhouse gases are sufficient as the only environmental indicator. Wouldn't endpoint indicators like Recipe do a better job?

Reply to Comment 4-3:

We agree that considering additional indicators would be a good avenue for future research. In this study, we wanted to focus on changes in agricultural production and consumption, economic welfare, greenhouse gas emissions and human health. We think that adding further impact indicators would stretch what can be succinctly described and explained in one paper. Following this and the last comment, we now mention the importance of other indicators in the Discussion section.

We would also like to note that our level of analysis is very much at the regional and country level, and not at the more detailed level at which LCA studies are conducted. In line with other food system analyses^{2,8}, our approach was to follow a processed-based analysis of environmental impacts that makes use of country-level statistics ascribed to specific activities and food groups. We appreciate that LCA methods such as ReCiPe represent complementing alternatives at different scales, including for more aggregated endpoints, and we now include a reference to ReCiPe in the Discussion section.

Comment 4-4:

The S-scenarios need more explanation. Are you really just redirecting land-based payments to your fruits, vegetables and nuts? Towards per-hectare payments for such crops? I find it disturbing that, this way, cereals are treated like meat, in spite of a very different environmental footprint.

Reply to Comment 4-4:

In line with the available statistics on how subsidies are provided, the payments as implemented in our model are based on land, labour, capital, inputs and outputs taking into account the situation in the specific sector and country. The treatment of subsidies therefore differs by food group: for example, land-based payments to the animal sector are generally very low or zero, whereas land-based payments for crops are fairly large (see Badri Narayanan: “Incorporating the Agricultural Domestic Support Data into GTAP 7 Data Base”). We describe this in the Supplementary Information as follows:

“...The tax variable t_0 is calibrated such that payments (PAY) based on outputs match the support estimates from the OECD database that are based on output. Similar formulas apply for the case of input and factor-use subsidies.”

The repurposing of payments was achieved by (i) decreasing domestic agricultural support based on land, labour, capital, inputs and outputs as present in the GTAP database, and (ii) providing output-based subsidies for horticultural products. We describe this procedure in more detail in the Supplementary Information in the section on “Representation of agriculture subsidies”.

On a more conceptual level, we do of course see the need for and importance of cereal production, and we agree that cereals have much lower environmental footprints than meat. However, our food-group approach, as explained in the Introduction, was intended to incentivise those food groups whose production, relative to projections, would have to increase under a transition towards healthier and more sustainable food systems. That meant to specifically focus on horticultural products as opposed to staple crops such as cereals.

Comment 4-5:

The manuscript is well written and clear.

Reply to Comment 4-5:

Thank you.

References

1. Balmford, A. *et al.* The environmental costs and benefits of high-yield farming. *Nature Sustainability* **1**, 477–485 (2018).
2. Springmann, M. *et al.* Options for keeping the food system within environmental limits. *Nature* **562**, 519–525 (2018).
3. Poore, J. & Nemecek, T. Reducing food’s environmental impacts through producers and consumers. *Science* **360**, 987–992 (2018).
4. Clark, M. & Tilman, D. Comparative analysis of environmental impacts of agricultural production systems, agricultural input efficiency, and food choice. *Environmental Research Letters* **12**, 064016 (2017).
5. Bechthold, A. *et al.* Food groups and risk of coronary heart disease, stroke and heart failure: A systematic review and dose-response meta-analysis of prospective studies. *Critical Reviews in Food Science and Nutrition* **59**, 1071–1090 (2019).
6. Willett, W. *et al.* Food in the Anthropocene: the EAT–Lancet Commission on healthy diets from sustainable food systems. *The Lancet* **393**, 447–492 (2019).
7. Huijbregts, M. A. *et al.* ReCiPe2016: a harmonised life cycle impact assessment method at midpoint and endpoint level. *The International Journal of Life Cycle Assessment* **22**, 138–147 (2017).
8. Laborde, D., Mamun, A., Martin, W., Piñeiro, V. & Vos, R. Agricultural subsidies and global greenhouse gas emissions. *Nat Commun* **12**, 2601 (2021).

Reviewer comments -

Reviewer #1 (Remarks to the Author):

I added additional information to the old review and replies of the authors.

Referee #2:

Comment 2-0:

A. Summary of the key results

Removing or changing agricultural subsidies may have important effects on diets. Simulations suggest that especially subsidies on fruits, vegetables and nuts have large effects on their production and consumption and therefore on health effects of diets. Abandonment of subsidies may therefore have negative health effects. Redirecting subsidies to fruits, vegetables and nuts gives positive health effects and climate benefits, but seems to have negative economic impacts. Both effects are straightforward. Equalising subsidies over countries based on population or GDP (keeping the same budget) increase global welfare. However, the countries who start with subsidies have a welfare loss. These welfare effects of subsidies (where effects on health are not included, as far as I understand) are a standard result of CGE analysis.

In order to do the analysis a health model and a general equilibrium economic model have been coupled and for environmental effects LCA results are used.

B. Originality and significance: if not novel, please include reference

As far as I know no one has done an exercise with respect to the relation between agricultural subsidies, health and greenhouse gasses. No one did couple a CGE model with health model and relating this with labour supply, as far as I know. However, I think that the general results are very straightforward. Subsidies imply a loss in equivalent variation (economic loss), but if they are targeted towards products that are better for health and climate, they give benefits on those terrains. This is not too surprising.

C. Data & methodology: validity of approach, quality of data, quality of presentation

A lot of effort is put in the exercise. However, in the main text only provides many numbers without an intuitive explanation of the mechanisms behind it and numbers which are highly uncertain.

D. Appropriate use of statistics and treatment of uncertainties

Uncertainties with respect to scenarios are included. As far as I see not the uncertainties with respect to parameters, which will be large.

F. Suggested improvements: experiments, data for possible revision

G. References: appropriate credit to previous work?

Yes

H. Clarity and context: lucidity of abstract/summary, appropriateness of abstract, introduction and conclusions

Abstract is clear. Introduction is very general. Conclusions are included in discussion.

GENERAL comments on the revision by reviewer 2.

Although some relevant additional material was presented, I still have difficulties in understanding the results, and it is impossible to calculate the results based on the materials presented.

Nevertheless, the rough results are quite intuitive.

Reply to Comment 2-0:

Thank you for your helpful comments and suggestions. In our revision, we have aimed to provide more detailed explanation of our methods and results. Please find a point-by-point response below.

Comment 2-1

Scenario descriptions could be more precise. For example, it could be explained in the main text that in scenario S100 all subsidies go into fruits, nuts and vegetables (as shown in table S9), (while in the benchmark about 20% of subsidies is allocated to horticultural products), instead of the abstract formulation "Repurposing of 25-100% of agricultural subsidies to support food commodities with beneficial health and environmental characteristics".

Reply to Comment 2-1:

Following the comment, we clarified what we mean with repurposing at the first instance. The results section on repurposing now reads as follows:

"Using agricultural subsidies (i.e. repurposing from previous ways of allocating subsidies) to support the production of foods with beneficial health and environmental characteristics led to ..."
REPLY Reviewer 2. It is just a choice not to make it more precise.

Comment 2-2:

It would also be useful to explain better how consumption is determined: what is the share of indirect consumption of agricultural products in total consumption? Also a list of demand elasticities would be useful, at least for the three main product categories. And added to that it would be useful to know the size of the subsidies as percentage of value added in order to get an idea of the price effect. For me it is surprising that the effect of subsidy removal from meat products is so much smaller than subsidy removal from horticulture, while the size of the sector is not that different and demand elasticities are probably higher in the meat related sectors.

Reply to Comment 2-2:

In our analysis, food consumption is determined by tracing primary production via intermediate sectors to the final consumer. For example, fruits and vegetables can be purchased directly from the primary production sector (denoted as "f&v" in our model aggregation), or for example as canned vegetables from the processed food sector (denoted as "ofd" for other processed food), or as part of a prepared meal in the restaurant sector ("sevcs"). Following the comment, we added this explanation to our Methods description and to the Supplementary Information.

As requested, we also added a table of the compensated demand elasticities we used (Table S1), and of price changes (Table S12). The reason that the effect of subsidy removal on meat is smaller than for horticulture is due to the fact that horticultural consumption is much larger in absolute (grams per day) terms than meat consumption, whilst the percentage reductions for both food categories are comparable. For example, global vegetable consumption is around 400 g/d while beef consumption is around 30 g/d. Following the comment, we added a table with baseline consumption values (in grams per day) to the Supplementary Information in support of this explanation (Table S5).

REPLY reviewer 2: useful that the tables are included.

Comment 2-3:

It is not clearly described how it is handled that not all production is consumed in the standard GTAP input-output tables; is all consumption allocated to the households by redefining the input-output tables? How are corrections made to make the percentage changes of consumption exactly the same as percentage changes of production?

Reply to Comment 2-3:

In general, the input-output tables and the CGE model structure assure that everything that is produced is also consumed in one way or another. However, not all agricultural products are consumed by the private household directly but could be further processed for non-food purposes like bioenergy or biochemical uses, which are also part of the model. In addition to representing intermediate processes and non-food uses, we also accounted for the amount of food that is wasted at the household level by using data from the FAO. We describe the latter in more detail in the Supplementary Information, and we added more detail on how intermediate sectors are represented in response to the previous comment.

REPLY reviewer 2: the added information is useful, but it doesn't solve the problem that part of agricultural production is not consumed directly, but through other sectors like ofd and services. Production

value of vegetables and fruits is about 1 trillion dollars, of which 60% is directly consumed and about 30% goes to ofd, beverages-tobacco and trade. As far as I know the ofd and service sectors have fixed parameters as inputs. In my opinion this should be mentioned somewhere.

Comment 2-4:

It is not clear how substitution of food is handled. In the CDE consumption function the cross price elasticities are low and the result of own price elasticities and income elasticities, while in practice an demand increase of horticultural products will be at the cost of some other food commodities. Information on percentage price changes of the different commodities as intermediate variable explaining the effect.

Reply to Comment 2-4:

Thank you for this comment. The substitution of food is a result of general equilibrium effects in our model. For instance, when subsidies are redirected to incentivise greater production of horticultural products, then less land and other production factors are available for other crops. This drives up the prices of those crops, which lowers their consumption. Following the comment, we expanded our explanation of such effects in the Methods section, and we appended an overview of changes in consumer prices in the Supplementary Information (Table S12). The revised Methods section now reads in this regard:

“The demand system in MAGNET resolves changes in food demand that are due to changes in the price of a specific food (as represented by own-price elasticities), in the price of other foods (as represented by cross-price elasticities), and due to changes in income (as represented by income elasticities). In MAGNET and other general-equilibrium models, household income is affected by changes in the agri-food system and by changes to other, non-agricultural sectors.”

REPLY reviewer 2: This doesn't solve the problem. Cross-price elasticities are about zero in the CDE demand functions, while in the main text, including the revision you give above, it is suggested that cross-price elasticities are in the model while it is not. This should be mentioned somewhere and it is certainly incorrect to suggest that they are in the model. Although the effect of the direct price elasticities may be such that the results are not too biased, there is no direct substitution method in the models.

Comment 2-5:

In the table below (made from the additional material, where explanation of the meaning of all abbreviations could be more complete) I see that the negative health effect of eating red meat is about half of the beneficial effect of eating more vegetables. [...] In the table below one explanation may be that the amount of fruits and vegetables eaten is very high: 549 gram per day. Relatively simple formulas must have been used that relate changes in consumption with deaths as a result of diet changes. It would be useful if they would be visible. I have some problem in interpreting the relative risk factors in table S5 in this context. I think it can be directly translated into avoided death per extra gram consumption. For lay people on this topic it would be informative to have these numbers.

Reply to Comment 2-5:

We used a comparative risk assessment for estimating the health impacts of changes in dietary risk factors. The assessment uses epidemiological data, including relative risk values that relate changes in risk factors (e.g. increased consumption of vegetables) to changes in the risk of dying from specific diseases. In addition, the calculations use disease-specific mortality rates and population numbers, each differentiated by age group and country. We provided the formula with which the change in diet-related disease burden is calculated in the Supplementary Information (see the section "Health analysis"). Following your comment, we also prepared an illustrative example that we discuss below.

Relative risk values are derived from epidemiological cohort studies (and meta-analyses of those) and describe the general dose-response relationship between a risk factor and a health outcome. As Table S7 (formaly S5) in the SI shows, a 100 g/d increase in red meat consumption is associated with an increase in the risk of dying from coronary heart disease (CHD) of 15% on average (RR=1.15, 95% confidence interval: 1.08-1.23), whereas a 100 g/d increase in vegetable consumption has been associated with a reduction in the risk of dying from CHD of 16% (RR=0.84,

95% confidence interval: 0.80-0.88).

What matters for calculating the diet-related disease burden is the change in risk exposure, i.e. the change (in grams per day) in vegetable and red-meat consumption (in line with the equation for calculating population impact fractions, PIFs, provided in the SI). In the EU, for example, vegetable consumption increased by 38 g/d under complete repurposing (see the change ("chg") of the waste-adjusted consumption parameter ("g/d_w") in the S100 scenario), whilst red meat consumption (which includes beef, lamb, pork) decreased by 3.4 g/d.

Multiplying the population impact fractions by mortality rates and population numbers (as described in the SI) indicates that the increase in vegetable consumption would be associated with 41,000 avoided deaths from CHD, and the reductions in red meat would be associated with 3,600 avoided deaths from CHD. Thus, the order of magnitude difference in the changes in consumption was roughly carried through to the final health estimates.

As described in the Supplementary Information, the exact calculations also account for other disease associations (Table S7), co-exposure to multiple risk factors (as death has to be ascribed to one specific risk factor in the assessment; see PIF equation), age-effects that attenuate risks at older ages, age-effects that differentiate mortality rates, as well as non-linear dose-response functions that reduce the impacts of an increase, e.g. in vegetable consumption, when baseline consumption is already relatively high.

REPLY reviewer 2: thank you for the clarification. I checked myself consumption levels, and was wrong with this. The example was very helpful. It made clear to me that I confused PIF and RR, although it is described in the table.

Comment 2-6:

In order to get the conclusions, you don't need much of the model, only the basic assumptions used. These are the current subsidies, an intuition on the order of magnitude of demand elasticities, and the supply elasticities. This could be more clearly formulated.

Reply to Comment 2-6:

We agree that examining a model's key parameters can provide a basic intuition on general trends and the orders of magnitude of interactions when all other things are held constant. This is especially the case in partial equilibrium analysis. However, in the type of general equilibrium analysis we did, everything depends on each other. For example, supply elasticities are not simply constant parameters but are complicated functions of the model's parameter space and functional forms. Similarly, the modelled health effects depend not only on relative risks and changes in exposure, but on a range of parameters, including the interactions handed down from the general equilibrium analysis. Following your comment, we clarified the description of our model framework and provided additional detail on the feedbacks and interactions represented by it. Please see in particular our replies to the two previous comments.

REPLY reviewer 2: Results of general equilibrium models have to a large extent the same type of results as partial equilibrium models. Let us do a short exercise for the POP scenario. Global vegetable and fruit production is about 1 trillion dollars, and the increase in global subsidies is about 180 billion (table S11), so 18%. However, the equilibrium price reduces with about 10% according to your results, implying that about half is accommodated through supply and half through demand. The price elasticity of demand is roughly 0.65 (taking the middle income one), resulting in an increase in demand of about 6.5 %, which is a little bit higher than your result that is about 4%, if I read figure S2 correctly. However, as mentioned before, about 1/3 of production is going through other sectors with roughly a price elasticity of demand of 0, so the effective price elasticity of demand in the model is about 0.4, that results roughly in your outcome. If we translate the result in grams per day, 4% of 587 gram (table S 5) is 23 gram p.p., and this seems to result in about 370000 avoided deaths (difficult for me to check).

Comment 2-7:

The effects of the subsidies on consumption and health are relatively uncertain. This could be stated more clearly.

Reply to Comment 2-7:

We followed established standards of reporting uncertainty related to health assessments. Our health analysis accounted for epidemiological uncertainty (see Table S7 in the Supplementary Information), and we reported the results as mean impacts including 95% confidence intervals. We explain this in the Supplementary Information as follows:

“For the different diet scenarios, we calculated uncertainty intervals associated with changes in mortality based on standard methods of error propagation and the confidence intervals of the relative risk parameters.”

To analyse the robustness of our results, including parameter-specific uncertainties, we also included a structured sensitivity analysis with different socio-economic pathways. We mention that in the Discussion section as follows:

“Our analysis was based on established systems models which are regularly used for policy assessments, and the results were robust with respect to socio-economic, environmental, and health-related uncertainties (Tables S13-S17).”

REPLY reviewer 2: for health effects this is correct. For the socio-economic effects parameter uncertainties are still not included, only variation in scenarios is included. Knowing that it is very difficult to give indications of parameter uncertainty in socio-economic models, it nevertheless should be mentioned that there is uncertainty.

Comment 2-8:

It is not clear to me why subsidies have such enormous effects on health effects of vegetables and fruits, and why abolition of meat subsidies has relatively small effects. The conclusion that abolishment of agricultural subsidies results in a reduction in health because of less consumption of horticultural products seems not very plausible to me. Global consumption of on average 600 grams of fruits and vegetables per person per day seems extremely high.

Reply to Comment 2-8:

Please see our response to Comment 2-5.

Regarding the consumption estimates: as explained in the Methods and Supplementary Information, we estimated baseline food consumption by adopting estimates of food availability from the FAO's food balance sheets, and adjusting those for the amount of food wasted at the point of consumption 1,2. Food balance sheets report on the amount of food that is available for human consumption 2. They reflect the quantities reaching the consumer, but do not include waste from both edible and inedible parts of the food commodity occurring in the household, which is why we adjusted those estimates for waste and processing.

An alternative would have been to rely on a set of consumption estimates that has been based on a variety of data sources, including dietary surveys, household budget and expenditure surveys, and food availability data 3,4. However, neither the exact combination of these data sources, nor the estimation model used to derive the data have been made publicly available. For some individual countries, using dietary surveys would also have been an alternative. However, underreporting is a persistent problem in dietary survey 5,6, and regional differences in survey methods would have meant that our results would not be comparable across countries.

In contrast to dietary surveys, waste-adjusted food-availability estimates indicate levels of energy intake per region that reflect differences in the prevalence of overweight and obesity across regions 7, which provides an additional test for general consistency.

REPLY reviewer 2. Thank you. I am convinced about the relatively large effects on vegetables and fruits.

Reviewer #2 (Remarks to the Author):

The authors analyse the impacts of agricultural subsidy changes on production, GHG emissions, consumption, health and economic welfare using an integrated economic-environmental-health

modelling framework on a global scale. The analysis demonstrates the importance of considering the feedbacks across these dimensions. The study is very valuable from both a scientific and an application-oriented point of view. The model used combines a general equilibrium model with a risk-disease model and environmental footprints of products and processes. The study can be a reference for complementary or further studies that consider the food system as a whole. The impacts of the scenarios of potential agricultural subsidy reforms on the different dimensions can also be used as a basis for decision-making. Therefore, from my point of view, this study is worth publishing.

Some minor points:

- It is unusual to place the (very well-founded and understandable) methodology at the end of the paper (but it is also possible in this way)
- Line 129 (or elsewhere): Perhaps specify if the countries without subsidies are a part of the non-OECD countries or if they build a separate group.
- Table S4 (supplement): The term in column "item" and the horizontal lines may be controlled.
- Table S11 (supplement): BMK=?

Reviewer #3 (Remarks to the Author):

Dear authors,

thank you for this interesting article. It contributes to a very relevant debate. It is crucial that food systems are considered as a whole by adopting a systemic view. I have that the MS was substantially improved following the comments of 4 reviewers. However, I have a few comments that have not or insufficiently been considered in the current version.

Major comments:

Comment 1a. The authors use the term "environment(al)", but in fact they include only part of the GHG emissions in their consideration.

All other environmental impacts are excluded, although there are now data available to cover at least part of these impacts. Impacts like water scarcity or losses of biodiversity are not really correlated to climate change. Therefore, I suggest to make this clear by replacing "environment/al" by "climate" or "climate change" or "GHG emissions" in the title, the abstract and in most parts of the text.

Comment 1b. Furthermore, the GHG emissions are limited to methane and nitrous oxide. Neither CO₂ from fossil fuels (such as fuels used in agriculture or in industrial processes for manufacturing agricultural means of production such as fertilizers) are included. It seems that land use change emissions (e.g. CO₂ from draining peat soils or deforestation are included). Please make this clear in the text. This makes the analysis rather incomplete and the term "environment" is even more misleading. The argument that the IPCC national GHG inventory methodology was followed is not valid here imho, since that methodology was developed to account for all emissions of a country (where the assignment to the sectors is of lower importance) and here you consider only the food sector. This fact should be made very clear in the main text; now it only becomes clear in the method section.

Comment 2. Figure 2 needs additional information:

Comment 2a. Missing/unclear units for 2b, 2d, 2e (should be per year)

Comment 2b. To assess the relevance of the changes, the changes in % would be useful in Fig. 2. You could enter them e.g. as numbers in the graph. Now they are available only in the suppl. mat.

Comment 2c. The different categories are not defined in the manuscript or supplement. Where are eggs? Where is seafood? What is in the "staples"? (or did I overlook something?) This information must be given; it could be added e.g. to Table S3.

Comment 2d. The level of aggregation in Fig. 2 and in the supplement is quite high. E.g. meat includes beef, pork and poultry, although from an GHG point of view (beef vs. poultry) and health impacts (red vs. white meat) they are very different. I understand that an aggregation is needed for a concise presentation, but a higher level of detail should be given in the supplement (e.g. in a table). We cannot expect the readers to reanalyse the raw data.

Comment 2e. Fig. 2a is presented in kg, which is questionable. Expressing production in calories or cereal units would make more sense.

Comment 3:

Staple crops are mentioned as detrimental for the environment and health. I would question that. First, staple crops are a very heterogeneous group (although it is unclear, what this group includes, see comment 2c). From an environmental perspective, staple crops are the most environmentally friendly to provide food energy, much more than fruits and vegetables. Furthermore, whole grains are listed among the positive factors for health, but not included in your analysis. The discussion should be more differentiated.

Minor comments:

* Lines 171-172, 202: give the unit for the number of deaths (per year?)

Dear reviewers,

Thank you for your comments and suggestions on our manuscript. Based on those we have further revised and clarified the manuscript. The following point-by-point response details the changes made in response to each of the reviewers' comments.

Responses to reviewers' comments:

Response to Reviewer #1:

Comment 1-0:

Although some relevant additional material was presented, I still have difficulties in understanding the results, and it is impossible to calculate the results based on the materials presented. Nevertheless, the rough results are quite intuitive.

Reply to Comment 1-0:

We very much appreciate your efforts in double-checking our results, and we are glad you were able to intuit the general direction of most aspects of our analysis. A closer replication of the results will be complicated by the fact that our model, in line with economic and epidemiological theories, contains a large set of non-linear equations, which is exactly why such numerical models were constructed (see, e.g., the MAGNET model description provided by Woltjer and colleagues, 2014). Whenever possible, we tried to provide intuitive explanations of our results, and based on your suggestions, we have added additional explanations to this end.

Comment 1-1:

Based on a clarification (albeit a small one) of how we describe the scenarios on repurposing subsidies ("Using agricultural subsidies (i.e. repurposing from previous ways of allocating subsidies) to support the production of foods with beneficial health and environmental characteristics led to ..."), you noted "It is just a choice not to make it more precise."

Reply to Comment 1-1:

We explain at multiple points throughout the manuscript what we mean with "foods with beneficial health and environmental characteristics" and also mention horticultural products in this specific sentence, together with noting how much we repurposed subsidies. The suggestion you had about mentioning the benchmark portion of subsidies devoted to those foods would suggest that those were already explicitly supported, which is not the case most of the time (rather they are used in those instances to grow horticultural products without subsidies specifically coupled to their production). We therefore didn't adopt your suggestions one-to-one, but added a reference to Figure 1 for that purpose. We hope our current formulation is clear enough considering that we describe the model scenarios at multiple points in the manuscript, including in the introduction, methods, and in parts of the results section.

Comment 1-2:

The added information [on how production is linked to consumption] is useful, but it doesn't solve the problem that part of agricultural production is not consumed directly, but through other sectors like ofd and services. Production value of vegetables and fruits is about 1 trillion dollars, of which 60% is directly consumed and about 30% goes to ofd, beverages-tobacco and trade. As far as I know the ofd and service sectors have fixed parameters as inputs. In my opinion this should be mentioned somewhere.

Response to Comment 1-2:

Following your comment, we now mention that the ofd and service sectors have fixed parameters as inputs. In particular, we state in the SI:

“The processed-food and service sectors use intermediate inputs in fixed proportions, which excludes substitution possibilities and therefore limits demand-side responses to changes in output.”

Comment 1-3:

Cross-price elasticities are about zero in the CDE demand functions, while in the main text, including the revision you give above, it is suggested that cross-price elasticities are in the model while it is not. This should be mentioned somewhere and it is certainly incorrect to suggest that they are in the model. Although the effect of the direct price elasticities may be such that the results are not too biased, there is no direct substitution method in the models.

Reply to Comment 1-3:

Following your comment, we removed our mentioning of cross-price elasticities from the Methods section and the SI.

Comment 1-4:

For the socio-economic effects parameter uncertainties are still not included, only variation in scenarios is included. Knowing that it is very difficult to give indications of parameter uncertainty in socio-economic models, it nevertheless should be mentioned that there is uncertainty.

Reply to Comment 1-4:

Following the comment, we now note in the Methods sections that we do not include parameter uncertainty:

“For a structured uncertainty analysis, we used different socio-economic development pathways that included more optimistic ones with greater economic and lower population growth, and more pessimistic ones with lower economic and higher population growth³². We did not include parameter uncertainty within these pathways.”

In addition, we changed our mentioning of socio-economic uncertainty to socio-economic variation in the Discussion section.

Response to Reviewer #2:

Comment 2-0:

The authors analyse the impacts of agricultural subsidy changes on production, GHG emissions, consumption, health and economic welfare using an integrated economic-environmental-health modelling framework on a global scale. The analysis demonstrates the importance of considering the feedbacks across these dimensions. The study is very valuable from both a scientific and an application-oriented point of view. The model used combines a general equilibrium model with a risk-disease model and environmental footprints of products and processes. The study can be a reference for complementary or further studies that consider the food system as a whole. The impacts of the scenarios of potential agricultural subsidy reforms on the different dimensions can also be used as a basis for decision-making. Therefore, from my point of view, this study is worth publishing.

Reply to Comment 2-0:

Thank you for your evaluation and your helpful comments and suggestions. Please find a point-by-point response below.

Comment 2-1:

It is unusual to place the (very well-founded and understandable) methodology at the end of the paper (but it is also possible in this way)

Reply to Comment 2-1:

We agree with that, but this ordering reflects the formatting guidelines for Nature journals. We provided a short summary of our methods at the end of the Introduction, which we hope makes the Results section clear enough.

Comment 2-2:

Line 129 (or elsewhere): Perhaps specify if the countries without subsidies are a part of the non-OECD countries or if they build a separate group.

Reply to Comment 2-2:

Following your comment, we re-formatted the table on our regional aggregation in the SI (Table S2) to clearly indicate which countries are grouped under OECD and non-OECD, respectively. We inserted a reference to the table in the main text and also mention the countries with the higher subsidy payments out of those groups in the preceding section on the current level of subsidies.

Comment 2-3:

Table S4 (supplement): The term in column "item" and the horizontal lines may be controlled.

Reply to Comment 2-3:

Following your comment, we redrew the horizontal lines in Table S4 (please note that the inner lines are meant to be dashed ones), and we clarified which items are listed under 'item' in the table's title as "Percentage of food wasted (wp) during consumption (cns), and percentage of processed utilisation (pctprcd)".

Comment 2-4:

Table S11 (supplement): BMK=?

Reply to Comment 2-4:

Thanks, BMK refers to benchmark/baseline. Following your comment, we inserted a short description of the scenario abbreviations in the table's title:

"BMK denotes the baseline (benchmark), and the reform scenarios include a conditioning of 25-100% (S25-S100) of subsidies to food commodities with beneficial health and environmental characteristics, and a combination of conditioning and regional restructuring in which each country provides subsidies in proportion to either its economy (GDP) or population (POP), whilst keeping the global amount of subsidy payments fixed."

Response to Reviewer #3:

Comment 3-0:

Thank you for this interesting article. It contributes to a very relevant debate. It is crucial that food systems are considered as a whole by adopting a systemic view. I have that the MS was substantially improved following the comments of 4 reviewers. However, I have a few comments that have not or insufficiently been considered in the current version.

Reply to Comment 3-0:

Thank you for your evaluation and your helpful comments and suggestions. Please find a point-by-point response below.

Comment 3-1:

The authors use the term "environment(al)", but in fact they include only part of the GHG emissions in their consideration. All other environmental impacts are excluded, although there are now data available to cover at least part of these impacts. Impacts like water scarcity or losses of biodiversity are not really correlated to climate change. Therefore, I suggest to make this clear by replacing "environment/al" by "climate" or "climate change" or "GHG emissions" in the title, the abstract and in most parts of the text.

Reply to Comment 3-1:

That's a good point and we agree. We initially included other footprints, but chose to focus on GHG emissions in the main text as those are most sensitive to supply and demand-side changes, whereas the other environmental impacts are relatively more modifiable by changes in farm-level management (e.g. changes in irrigation, nutrient recycling, etc).

Following your comment, we replaced "environmental" with "climate" or "climate change" in the title and also in the text when this could otherwise be misleading. In the Discussion section, we have included a dedicated caveat that is related to our focus on GHG emissions and climate change:

"Further work is also needed to quantify the implications of subsidy reform on other environmental dimensions, including water and pesticide use which some horticultural products do not currently perform well on ^{13,14,17}."

Comment 3-2:

Furthermore, the GHG emissions are limited to methane and nitrous oxide. Neither CO₂ from fossil fuels (such as fuels used in agriculture or in industrial processes for manufacturing agricultural means of production such as fertilizers) are included. It seems that land use change emissions (e.g. CO₂ from draining peat soils or deforestation are included). Please make this clear in the text. This makes the analysis rather incomplete and the term "environment" is even more misleading. The argument that the IPCC national GHG inventory methodology was followed is not valid here imho, since that methodology was developed to account for all emissions of a country (where the assignment to the sectors is of lower importance) and here you consider only the food sector. This fact should be made very clear in the main text; now it only becomes clear in the method section.

Reply to Comment 3-2:

Following your comment, we now mention our focus in methane and nitrous oxide in the methods summary of the Introduction:

“In our environmental analysis, we focus on changes in agricultural greenhouse gas emissions (specifically methane and nitrous oxide) because greenhouse gas emissions, compared to other environmental impacts, are less modifiable by farm-level management and more by changes in the mix of production ⁵.”

Comment 3-3:

Figure 2 needs additional information:

- (a) Missing/unclear units for 2b, 2d, 2e (should be per year)
- (b) To assess the relevance of the changes, the changes in % would be useful in Fig. 2. You could enter them e.g. as numbers in the graph. Now they are available only in the suppl. mat.
- (c) The different categories are not defined in the manuscript or supplement. Where are eggs? Where is seafood? What is in the "staples"? (or did I overlook something?) This information must be given; it could be added e.g. to Table S3.
- (d) The level of aggregation in Fig. 2 and in the supplement is quite high. E.g. meat includes beef, pork and poultry, although from an GHG point of view (beef vs. poultry) and health impacts (red vs. white meat) they are very different. I understand that an aggregation is needed for a concise presentation, but a higher level of detail should be given in the supplement (e.g. in a table). We cannot expect the readers to reanalyse the raw data.
- (e) Fig. 2a is presented in kg, which is questionable. Expressing production in calories or cereal units would make more sense.

Reply to Comment 3-3:

Thank you for these helpful comments.

Re (a): all impacts are for scenario impacts in the year 2030 relative to a benchmark scenario in 2030 that does not contain the scenario-related changes. We now clarified that in the figure description as “Impacts are for the year 2030 compared to a business-as-usual scenario without reforms”.

Re (b): We thought about displaying the percentage changes in this figure, but because our focus was on global subsidy reform, we preferred displaying absolute changes as those sum to the global ones. In contrast, percentage changes might look small or large without providing this kind of contextual information. We think inserting percentage changes in addition to the absolute ones would make the figure too cluttered, but we provide them in the main text and in the appendix.

Re (c): Thanks, we added this information in the description of the figure as “Food groups include wheat, and other cereals and grains (staples), vegetables, fruits, and other horticultural products (fruits&veg), vegetable oils and sugar (oil&sugar), beef, lamb, pork, and poultry (meat), milk and dairy products (milk), and other food commodities (other).”

Re (d): We tested different levels of aggregation for the figures, and the level of aggregation was chosen such that it suited the level of change in the scenarios and did not become too cluttered. The same reasoning applies to the display items in the SI. However, the supplementary data file includes both an aggregate and a detailed resolution. We formatted

the data as a pivot table, so no re-analysis of raw data is needed for switching between the two resolutions.

Re (e): We chose to use weight as a production unit, because production is commonly expressed in weight, and because it is related to the environmental and health analyses: environmental footprints are commonly expressed as impact per weight (e.g. per kg), and changes in diet-related risks are measured in changes in servings measured in weight.

Comment 3-4:

Staple crops are mentioned as detrimental for the environment and health. I would question that. First, staple crops are a very heterogeneous group (although it is unclear, what this group includes, see comment 2c). From an environmental perspective, staple crops are the most environmentally friendly to provide food energy, much more than fruits and vegetables. Furthermore, whole grains are listed among the positive factors for health, but not included in your analysis. The discussion should be more differentiated.

Reply to Comment 3-4:

We very much agree that staple crops are not detrimental for the environment or health, and we did not mean to create that impression. We mention staple crops in the Introduction in the context of a demand analysis as follows:

“... instead of additional global increases in the production of staple crops, animal-source foods, and sugar crops – estimated at 40-80% between 2010 and 2050 – a food system underpinning healthy and sustainable diets would require shifts from those food groups to foods that are both healthy and lower in environmental resource use and pollution, such as fruits, vegetables, legumes, and nuts and seeds.”

In this sentence, we grouped staple crops together with animal source foods and sugar crops, not because staple crops are unhealthy and unsustainable, but because there is no shortage in production relative to how much a well-balanced healthy and sustainable diet should contain. Demand would also go down under healthy and sustainable diets because less staple crops would need to be grown for use as animal feed. This is in contrast to fruits, vegetables, legumes, and nuts and seeds, all of which would have to be produced at much higher quantities for everybody to have the opportunity to have a healthy diet, making them a prudent target for receiving subsidies. As our analysis shows, there is no danger of shortages in the production of staple crops, even if subsidies were to be redirected.

Comment 3-5:

Lines 171-172, 202: give the unit for the number of deaths (per year?)

Reply to Comment 3-5:

Yes, it's per year, as are the other impacts. Following your comment, we inserted “in 2030” after each mention of mortality impacts to make clear that the number of deaths refers to that specific year.

Reviewer comments, further –

Reviewer #2 (Remarks to the Author):

The manuscript has been revised carefully. My few comments have been considered, especially the indication which country belong to which country group and the clarifications of some tables. (Considering the ordering of the article sections – methodology section at the end – I have realized the difference to other journals since I am actually involved in a Nature Sustainability article.) Furthermore, the specifications induced by the qualified comments of the other reviewers have further increased the accuracy and comprehensibility of the manuscript. The study is well documented and broad-based.

The comprehensive approach of considering the whole food system from production to consumption including the different dimensions and the feedbacks across these dimensions is becoming increasingly important in science and in policy implementation. The present study can further support these efforts and provide a basis for the scientific discussion. Although the aggregation level of the study is high and further analysis will be required, the study is worth publishing from my point of view.

Reviewer #3 (Remarks to the Author):

Dear authors,

thank you for the revised manuscript. My comments were taken into account satisfactorily and I recommend to accept this manuscript.